# Real-Time Implementation of a Microcontroller-Based Coupled-Tank Water Level Control System with Feedback Linearization and Fuzzy Logic Controller Algorithms

**DOI:** 10.3390/s25051279

**Published:** 2025-02-20

**Authors:** Bahadir Yesil, Savas Sahin

**Affiliations:** 1Baylan Water Meters Research and Development, Izmir 35620, Türkiye; 2Electrical and Electronics Engineering, Faculty of Engineering and Architecture, Izmir Katip Celebi University, Izmir 35620, Türkiye

**Keywords:** nonlinear control, feedback linearization, fuzzy logic controller, liquid level control, microcontroller, coupled tank system

## Abstract

An embedded coupled tank system (CTS) based on an ARM Cortex M7 microcontroller is developed for liquid level control, utilizing three nonlinear control strategies: fuzzy logic controller (FLC), feedback linearization (FL)-based proportional integral derivative (PID) controller, and FL-based FLC. The goal is to maintain an accurate predefined water level in the second tank and compare the performance of these controllers in terms of overshoot, settling time, and tracking accuracy. The CTS model is developed using physical parameters obtained from a real experimental setup. The controllers are first simulated in Matlab/Simulink and then tested on a physical CTS prototype, featuring a microcontroller interface with pumps, actuators, and sensors for real-time control and data acquisition. A custom graphical user interface and software are developed for conducting experiments and acquiring data. Results show that FL-based FLC and FL-based PID controllers offer better overshoot and tracking performances than FLC on nonlinear CTS. However, FLC and FL-based FLC outperform FL-based PID in settling time, with FL-based FLC showing very good performance in terms of tracking accuracy, overshoot, and settling time.

## 1. Introduction

The coupled tank system (CTS) is an electromechanical system consisting of level sensors, an electrical control circuit, a water pump, and at least two tanks interconnected to each other [1,2,3,4,5,6,7]. In industrial applications, CTSs are widely used for liquid level regulation to maintain steady flow rates and prevent overflow or underflow conditions [8,9,10,11]. Such systems are commonly implemented in water treatment plants, where precise liquid level control is crucial for efficient filtration and chemical dosing. Additionally, CTS is used in petrochemical industries to regulate fuel and chemical storage levels, ensuring safety and optimal process efficiency. These applications highlight the importance of advanced control strategies for maintaining accurate liquid levels in real-world industrial environments. In academic applications, it serves as a benchmark experimental setup for engineering education and research on the related control systems and fluid dynamics [12,13,14,15]. CTS tank level is measured via ultrasonic sensors in real time, which offers good performance features such as robustness, low cost, and ability to be contactless [16,17]. In the related literature, an ultrasonic level sensor is developed with a low-cost Arduino electronics card [18]. An ATMega 8535 Microcontroller (MCU) is employed to acquire the liquid level by using ultrasonic signals [19]. Similarly, a level measurement module is developed with an ultrasonic sensor. It has a standard deviation of 0.71756 cm in the range of 5 cm to 150 cm [20]. In another design, a fuzzy logic system is implemented on an Arduino Uno electronics card [21]. MCU-based controller designs are widely implemented for physical water level control systems by researchers owing to their low cost and stand-alone capability [22,23,24,25,26].

As for the proportional integral derivative (PID) controller design for CTS, a generalized PID controller is developed and compared with the conventional PID controller performance [27]. In another study, the tuning methods of a conventional PID controller were evaluated in a CTS simulated environment in terms of rise time, settling time, steady-state error, and overshoot [28]. Ziegler–Nichols (ZN) and Root–Locus method-based PID controller designs were compared for CTS level control [29]. The conventional PID controller is tuned by auto-tuning, ZN, and Tyreus–Luyben for a single tank level control [30]. In another application, a proportional integral (PI) controller was performed for CTS by using ZN, Ciancone correlation, and pole placement tuning methods, and controller performances were compared [31]. A modified PID controller was applied to the simulated CTS platform for rise time, settling time, and overshoot [32].

When it comes to fuzzy logic controllers (FLC) for CTS, a developed Mamdani-type FLC was realized for minimum overshoot and steady-state error [33]. A developed Sugeno-type FLC was utilized to determine the tank pump speed with the tank water level and light intensity [34]. In another application, a Sugeno-type FLC algorithm was embedded in the MCU controlling the tank water level [35]. A Sugeno-type FLC embedded programmable logic controller was used for controlling liquid level [36]. The developed mathematical model of CTS was obtained from real system data, and it was used for designing the self-tuning fuzzy PID controller [37].

Feedback linearization (FL) is widely known as a powerful nonlinear control strategy for CTS. A CTS model having inherent nonlinearities was implemented in an FL strategy to be controlled by controllers [38,39,40]. A FL-based linear discrete time controller was used on a considered plant [41]. FL-based advanced nonlinear controller algorithms were developed for similar CTS plants [42,43,44].

In this study, the FLC, FL-based PID controller, and FL-based FLC, which are designed as nonlinear controllers, are implemented in an embedded MCU with C codes for the level control of a physical CTS without the need for the external control program Matlab/Simulink. Herein, the ARM Cortex M7 MCU, offering superior processing capabilities, acquires liquid flow data and level sensor data and controls actuators like valves and motors. They are implemented on the MCU via non-blocking code architectures, direct memory access (DMA), and interrupts [45,46]. The embedded software is used with the Keil MDK–ARM environment. Hardware schematics of the control circuit are prepared to interface the MCU with a CTS having sensors, valves, and a water pump. The developed graphical user interface (GUI) designed in C# using MS Visual Studio Community 2022 is used for real-time controller tests. A serial communication protocol is used for data communication between the MCU and the GUI. CTS modeling and simulations are tested in the Matlab/Simulink environment. CTS experimental results are conducted for various references in real-time test conditions. The simulation and experimental results of the developed controllers are compared to each other in terms of overshoot, settling time, and tracking error. The performance metrics are determined as root-mean-square error (RMSE), mean absolute error (MAE), and mean square error (MSE).

The remaining parts of the study are structured as follows. Section 2 details the materials and methods employed to implement the CTS, covering both its mechanical and electrical construction. In Section 3, the simulation model of the CTS is developed. This includes the description of the FLC, FL-based PID controller, and FL-based FLC within simulation blocks. Results from both simulation and real-time controllers are presented, along with a comparative analysis of the performance of the candidate control methods. Section 4 outlines our conclusions and projections on CTS candidate controller strategies.

## 2. Materials and Methods

The CTS model covers two physically connected tanks linked by pipes, valves, and a common reservoir where one tank serves as the source while the other acts as the destination [1,2,3,4,5,6,7,47]. In the electromechanical setup of the CTS, it has the following components: (i) a water pump, (ii) two electromagnetic flow meters, (iii) two ultrasonic water level sensors, and (iv) servo motor-controlled ball valves as adjustable flow orifices (Figure 1). In CTS operation, water continuously circulates by flowing between two tanks and the reservoir. The water levels in each tank are defined as state variables, while the flow rate is regulated by the pressure differential between the tanks. A comprehensive explanation of the CTS model and its components, along with the controller design, are represented in the sub-sections.

### 2.1. State Space Representation of CTS

$Q_{in},\;Q_{mid},$ and $Q_{out}$ represent the flow entering the cylindrical CTS, the flow between the two tanks, and the total flow exiting the system, respectively [7,18]. Specifically, $Q_{in}$ corresponds to inlet flow $Q_{1}$, as shown in Figure 1, while $Q_{mid}$ represents $Q_{2}$. In the flow model of CTS, $Q_{out}$ is the outlet flow equal to the sum of $Q_{3}\;and\;Q_{4}$, which drain from the second tank. The mathematical model is given in Equation (1).(1)$$\dot{x_{1}}=\frac{{dh}_{2}}{dt}=-\frac{\left(s_{2}a+s_{3}a\right)\sqrt{2g}}{A_{2}}\sqrt{x_{1}}+\frac{s_{1}a\sqrt{2g}}{A_{2}}\sqrt{x_{2}}\;\dot{x_{2}}=\frac{\left(s_{2}a+s_{3}a\right)\sqrt{2g}}{A_{2}}\sqrt{x_{1}}-\frac{\left(A_{1}+A_{2}\right)s_{1}a\sqrt{2g}}{A_{1}A_{2}}\sqrt{x_{2}}+\frac{k_{PWM}Q_{max}}{A_{1}}\;y=h_{2}=x_{1}$$
where $A_{1}$ and $A_{2}$ represent the surface areas of the tanks, and a denotes the cross-sectional area of the pipe with $s_{n}$ valve opening ratio. Defining state variables $x_{1}=h_{2}$ and $\;x_{2}=h_{1}-h_{2}$, the dynamic model of the coupled tank system is obtained.

### 2.2. Layout of Implemented CTS

The implemented CTS system, as illustrated in Figure 2, uses a USB-serial port interface to connect the ARM Cortex-M7 MCU-based controller to a computer with a GUI, developed using C# within the MS Visual Studio Community 2022 on the Dot.Net platform [48]. The custom-designed interface circuit includes drivers and level shifters for controlling sensors and actuators used in the CTS. The system is powered by two DC power supply modules: one with single-output DC24V 2A, and the other with dual-output DC12V-5V, each capable of supplying a 5A current. CTS also includes two electromagnetic flow meters for measuring input and output flow rates, two ultrasonic level sensors for tank 1 and tank 2 (with 3D printed enclosures), and three servo motor-controlled ball valves to manage flow between the tanks, output flow, and drain (by-pass) flow. Transparent pipe connections between the tanks, inlet, and outlet sections allow for visual monitoring of the flow and verifying that no disturbances or air bubbles occur during experiments. Two cylindrical transparent tanks (each 60 cm in height and 9 cm in outer diameter) are used in CTS, with a main water reservoir (60 cm × 25 cm × 25 cm) for storing the total volume of water used in experiments. A submersible DC12V (JT-500 model, manufacturer Shenzhen iSmart Electronic, Shenzhen, China) 600 l/h capacity water pump, which is used for supplying inlet flow into the first tank, is assembled in the water reservoir. A full H-bridge circuit is used to control the water pump motor with a maximum current protection of 4 A.

The MCU-based development board is equipped with a 320 × 240 resolution display, used for real-time display of controller status and parameters. The development board facilitates the implementation of real-time control algorithms by the pure computing capability of the internal ARM Cortex-M7 MCU(STM32F746NG from ST Microelectronics, Geneva, Switzerland), without any external computer hardware/software support. GUI is used for collecting real-time data from the MCU, which are then transferred to Matlab for calculating and comparing the controller performances. In case of a tank overflow or electrical malfunction, an emergency stop button is included to shut down the CTS system. An electrical block diagram of the CTS is depicted in Figure 3.

### 2.3. Sensors and Actuators of Implemented CTS

#### 2.3.1. Level Sensors

Ultrasonic sensors used in the CTS real experimental setup utilize sound waves to measure the distance to the fluid surface [18,19]. An ultrasonic sensor type is used in this coupled tank setup for calculating water levels due to its characteristic advantages, including non-contact measurement capability and linearly wide measurement range without the need for run time calibration [18,21,22,23]. The cover of the 3D-printed case, in which the HC-SR04 (Fbele Electronics, Ningbo, China) sensor is mounted, features specially designed moisture evacuation holes to prevent the remaining water in the tank from evaporating and damaging the sensors (Figure 4a,b).

The ultrasonic level sensor is triggered by the MCU with a 10 μs pulse, which starts the measurement cycle of the sensor. After that, the sensor controller transmits a pulse train of 8 pulses at 40 kHz frequency to the transmitter, which emits ultrasonic waves. Ultrasonic waves travel down through the air from the upper empty part of the tank, hit the water surface, and then bounce back to the sensor upwards where the receiver section detects the reflected waves. The pulse width of the resulting echo signal is measured by the internal timer of the MCU to determine the water level in the tank.

#### 2.3.2. Electromagnetic Flow Meters

There are two electromagnetic flow meters used in the design of the implemented CTS setup. The output voltage is proportionally dependent on the velocity of the liquid, and hence can be used to observe the flow rate. The flow rate, which is visible on the display, is also obtained by the frequency output at the AF-E-400 flow meter (Krohne Messtechnik GmbH, Duisburg, Germany) in Figure 5.

#### 2.3.3. Servo Controlled Ball Valves

The servo-controlled ball valve regulates the flow of water through the valve (Figure 6a,b). The motor is connected to the valve stem, which has been specifically designed and precisely 3D-printed as part of this study. Once the motor runs, the water passes through the valve in a controlled manner. The MG-996R servo motor (Tower Pro Pte Ltd., Singapore) is used in the CTS setup. It is capable of delivering up to 11 kg-cm of torque to provide a high degree of precision and force, while it can operate in a wide voltage range from 4.8 V to 7.2 V.

The electrical schematic of the servo motor is given in Figure 7, where the AA51880 integrated circuit (IC,Agamem Microelectronics, Hsinchu, Taiwan) provides the drive for the servo motor and uses external N and P-Channel MOSFET transistors for the driver output [49]. The control input for the servo position is the variable duration pulse in the range of 1–2 ms, in a modulation period of constant 20 ms. The shortest pulse width of 1 ms corresponds to full left (anti-clockwise), while 2 ms is full right (clockwise), and 1.5 ms is the center (neutral) position. The 5K Ohm feedback potentiometer is attached to the output shaft, which is geared down from the motor for increased torque and lowered operational speed. The potentiometer is only used for 90° (half) of its range, which corresponds to a full open-close cycle of the assembled ball valve (from full-left to center position only). The internal structure of the driver IC provides pulse-width decoding for the input pulse width modulation (PWM) signal, where the pulse width is converted into a control voltage to be compared with the output voltage of the feedback potentiometer. The internal IC design also generates a dead band to prevent jittering.

### 2.4. MCU Electronics Card of CTS

The embedded hardware design consists of a main controller unit, actuators, and sensors used for observing and controlling the CTS setup. The MCU-based electronic board serves as the primary control and processing unit [45]. It gathers all data from the environment, performs control algorithms, and manages the flow of data between the PC and other components. The implemented electronic interface card can be used as a data acquisition control card where physical connections can be seen in Figure 8.

### 2.5. Embedded Software Flowchart and Control Algorithms Implemented in C Language

The embedded software is developed via C language within a Keil MDK-ARM™ environment V5.31 (Figure 9). The main software gathers level and flow measurements. It uses serial communication for the GUI by using a non-blocking approach, DMA, and interrupts. The determined timer of 40 ms triggers ultrasonic level measurements with five sequential datasets with a median filter. The level can thus be calculated every 200 ms. This interval also serves as the timing base for the developed FLC, FL-based PID, and FL-based FLC controllers. Flow rates, tank levels, control signals, and communication with the PC can be completed in a period of 1500 cycles (300 s). System input and output variables are displayed on screen within 1 s.

### 2.6. PC Based GUI Software Design

The GUI is developed with MS Visual Studio Community 2022 integrated development environment. The serial communication is initialized, and the visual layout of the CTS is designed within Windows Forms. The developed GUI allows switching between FLC, FL-based PID, and FL-based FLC. Additionally, a manual mode is incorporated for the hardware diagnostics and tests. Pump speed, tank levels, flow rates, and valve positions are updated every second on the GUI (Figure 10). An open-source analog gauge component [50] is utilized to display the flow rate on the panel.

Real-time test data are transferred to the GUI through the serial port during the tests. Each test cycle is defined as 300 s. In the GUI screen, a “rich text box” component is cleared, and a first line is added with a date-time stamp and control mode in the beginning of the test cycle. The text box is appended by the serial data received from the MCU board after every sampling period within the ongoing test cycle. With a sampling period of 200 ms, 1500 data for each variable are collected during a single test cycle of 300 s. Each sampling period yields a line in the rich text box, with comma separated values of “sampling time in seconds, first tank level, second tank level, set value for second tank level, input flow rate, output flow rate, input pump PWM percentage”. Suffixes CR and LF are added to continue in another line for the next set of sampling data. After the test time is over, the water pump is stopped, and the drain valve is opened for quick discharge of tanks to get ready for the next test cycles. At the same time, the rich text box contents are appended by a single line including the information “END OF LOG FILE” with a date and time stamp and the valve opening ratios—cavities during the test. Then, the entire contents of the rich text box are copied in an “rtf” file by using the Visual Studio “RichTextBox.SaveFile” method. Data in these files in CSV format are imported to Matlab R2022b for controller performance calculations.

In the physical layer of the developed system, an efficient protocol is developed to manage communication between the MCU and the PC. In this protocol, R, W, and C are used to denote read, write, and control commands, respectively. Sensors and actuators are defined as L for level sensor, V for valve, and P for pump. All possible commands with their parameters are listed in Table 1.

### 2.7. FLC Design

A two-input and single-output Mamdani-type FLC is designed for the CTS. One of the inputs is defined as the error between the desired and actual $h_{2\;}$ level of the water in the second tank of the CTS, while the other input is defined as the change in error. Five linguistic variables used for fuzzification of “error” and “change in error” inputs are given in Figure 11a as Negative Large (NL), Negative Small (NS), Zero (Z), Positive Small (PS), and Positive Large (PL). The output of the designed FLC is defined as the PWM duty cycle of the DC motor, which pumps water into the first tank of the CTS. Three linguistic variables defined for fuzzy output are High (H), Medium (M), and Low (L), which are given in Figure 11b.

Triangular and trapezoidal membership functions are characterized by piecewise linear parts. The designed sub-program allows for computation of the membership degree, effectively reducing the code size of the FLC (Figure 12). The “IF-THEN” rules are defined in Table 2. In C programming, each rule is defined as a structure with three unsigned integer variables. The Mamdani max-min inference method is used for code efficiency, and all 25 rules are evaluated in a single software loop in the rule evaluation subroutine. As for the defuzzification subroutine, the center of gravity (COG) method is implemented in the code. A defuzzification algorithm calculates the COG from the membership function distribution in 100 iterations/steps for integrals. In other words, the total area under the membership functions is divided into 100 sub-areas to calculate the COG of the combined control action. This division is performed to discretize the integration process into smaller steps for numerical calculation. All FLC flow charts are given in Figure 13.

To ensure that the FLC works within the physical constraints of the system, quantification factors are calculated based on the physical limits of the CTS and experimental trials. Error quantification is determined by physical limits related to the height of the tank. Since the error is the difference between the desired and actual water levels, the error boundaries are set to reflect practical values. These boundaries are calculated by selecting a range close to the tank’s height, with an error range of approximately ±60 cm from the desired level. This provides the FLC with a clear understanding of how far off the tank is from the desired set point. Error rate quantification (change of error) is more complex, as it takes into account the dynamic behavior of the tank system. The quantification is based on two extreme operational conditions: (i) Maximum positive change of error: This occurs when the flow rate into the tank is at its maximum, and the drain is zero, resulting in the largest possible positive change in error. This scenario corresponds to a situation where the tank is rapidly filling. (ii) Maximum negative change of error: This happens when there is no inflow into the tank, and the drain is at its maximum, causing the water level to decrease rapidly. This condition typically occurs when the tank reaches its maximum level, and the water starts to drain quickly. The quantification factors for both error and error rate are experimentally determined. The error quantification factor is set to 40, and the error rate quantification factor is set to 20, based on these boundary conditions and the CTS’s behavior. The output of the FLC is the crisp control signal for the pump, and it is normalized by calculating the quantification factor that incorporates the dead zone of the pump’s PWM control signal as 0.8. The dead zone corresponds to the ineffective range of PWM from 0% to 20%, where the motor does not rotate or provide sufficient flow. This output normalization process ensures that the FLC output remains within the physical limits of the pump and provides smoother control.

### 2.8. FL and FL Based CTS Model

FL is a widely studied and extensively applied control strategy in the literature for managing nonlinear dynamical systems [38,39,40,41,42,51]. The primary objective of FL is to transform a nonlinear system into an equivalent linear form by employing suitable state-space transformations and mathematical axioms. This transformation allows for the elimination of system nonlinearities, thereby facilitating the application of conventional linear control techniques. By utilizing FL, the equilibrium points of a nonlinear system can be effectively linearized, ensuring that the overall closed-loop system behaves in a linear manner across a broad range of operating conditions. One of the key advantages of FL is its applicability to single-input single-output (SISO) nonlinear systems that satisfy certain differentiability and controllability conditions [43,44,51,52]. In practical implementations, control objectives are often centered on achieving precise tracking of output variables while maintaining system stability and robustness. To address these challenges, input-output feedback linearization is commonly used, which involves systematically canceling out the nonlinearities of the system through a well-defined coordinate transformation. This transformation effectively maps the original nonlinear dynamics into a feedback linearized representation, thereby enabling the design of controllers based on control methodologies. The importance of FL extends to various real-world engineering applications, particularly in fields such as fluid dynamics where nonlinear effects often pose significant control challenges. In the outer loop, controller strategies may struggle to handle these nonlinearities effectively, leading to suboptimal performance. In such cases, FL provides a systematic approach to overcoming these limitations by ensuring that the nonlinear system behaves in a predictable and well-regulated manner. In this study, FL is applied to a CTS, which inherently exhibits nonlinear flow dynamics due to the interaction between liquid levels, flow rates, and actuator responses. The primary goal of implementing FL in this context is to facilitate effective water level regulation by transforming the nonlinear CTS model into a linear domain. This transformation enables the application of structured control strategies that significantly improve performance in terms of tracking accuracy, overshoot reduction, and transient response characteristics. By leveraging input-output feedback linearization, the CTS is converted into an approximately linear system, allowing for the design and implementation of advanced controllers that enhance both stability and robustness. The input-output FL of CTS can be derived by using Equation (1). The obtained new input v as a linearized input for the nonlinear system can be expressed in Equation (2) as follows:(2)$$\dot{y}=\dot{x_{1}}=\frac{{dh}_{2}}{dt}={-c}_{1}\sqrt{x_{1}}+c_{2}\sqrt{x_{2}}\;\ddot{y}=\ddot{x_{1}}=-\frac{c_{1}}{2}\frac{1}{\sqrt{x_{1}}}\dot{x_{1}}+\frac{c_{2}}{2}\frac{1}{\sqrt{x_{2}}}\dot{x_{2}}\;\dot{x_{2}}=c_{1}\sqrt{x_{1}}-{2c}_{2}\sqrt{x_{2}}+\frac{u}{A}\;\ddot{y}=-\frac{c_{1}}{2}\frac{1}{\sqrt{x_{1}}}\left({-c}_{1}\sqrt{x_{1}}+c_{2}\sqrt{x_{2}}\right)+\frac{c_{2}}{2}\frac{1}{\sqrt{x_{2}}}\left(c_{1}\sqrt{x_{1}}-{2c}_{2}\sqrt{x_{2}}+\frac{u}{A}\right)\;\ddot{y}=\frac{c_{1}^{2}}{2}-\frac{c_{1}c_{2}}{2}\frac{\sqrt{x_{2}}}{\sqrt{x_{1}}}+\frac{c_{1}c_{2}}{2}\frac{\sqrt{x_{1}}}{\sqrt{x_{2}}}-c_{2}^{2}+\frac{c_{2}}{2A\sqrt{x_{2}}}u\;\dot{z_{1}}=z_{2},\;\dot{z_{2}}=v,\;y=z_{1}\;v=f\left(x\right)+g\left(x\right)u\;f\left(x\right)=\frac{c_{1}c_{2}}{2}\left(\frac{\sqrt{x_{1}}}{\sqrt{x_{2}}}-\frac{\sqrt{x_{2}}}{\sqrt{x_{1}}}\right)+\frac{c_{1}^{2}}{2}-c_{2}^{2}\;g\left(x\right)=\frac{c_{2}}{2A\sqrt{x_{2}}}$$

### 2.9. FL-Based PID Controller

The FL technique is used for transforming CTS into a linear system model by Equation (2). The nonlinear system input u can be written with the linear input v in Equation (3).(3)$$u=\frac{1}{g(x)}\left\{-f\left(x\right)+v\right\}\;u=\frac{2A\sqrt{x_{2}}}{c_{2}}\left\{-\frac{c_{1}c_{2}}{2}\left(\frac{\sqrt{x_{1}}}{\sqrt{x_{2}}}-\frac{\sqrt{x_{2}}}{\sqrt{x_{1}}}\right)-\frac{c_{1}^{2}}{2}+c_{2}^{2}+v\right\}$$
where $c_{1}$ and $c_{2}$ constants are calculated by using physical parameters of designed CTS: (i) tank surface area $\left(A\right)\;5411\;{mm}^{2}$, (ii) orifice between tanks $\left(a_{2}\right)\;66.48\;{mm}^{2}$, (iii) flow output orifice $\left(a_{3}\right)\;22.28\;{mm}^{2}$, (iv) fast drain orifice $\left(a_{4}\right)\;66.48\;{mm}^{2}$, (v) flow coefficients $\left(s_{2},\;s_{3}\right)\;1$, (vi) drain flow coefficient ${(s}_{4})\;0$, and (vii) gravity $\left(G\right)9.8066\;m/s^{2}.$(4)$$c_{1}=\frac{(s_{3}a_{3}+s_{4}a_{4})\sqrt{2g}}{A}=0.01824\;c_{2}=\frac{s_{2}a_{2}\sqrt{2g}}{A}=0.05441\;u=0.1989\sqrt{x_{2}}\left\{-0.0004961\left(\frac{\sqrt{x_{1}}}{\sqrt{x_{2}}}-\frac{\sqrt{x_{2}}}{\sqrt{x_{1}}}\right)+0.002794+v\right\}$$

In Equation (4), the linear input might be controlled by the PID controller on the FL-based CTS. The PID controller parameters are determined using Matlab R2022b’s system identification tool [53]. Herein, input and output data pairs are used to find the discrete-time transfer function of the CTS, and it is derived as $\frac{0.534z^{-1}}{1-1.963z^{-1}+0.9632z^{-2}}$. With the sampling time of 0.2 s, the tuned PID gain parameters are obtained as 0.4031, 1.1128, and $1.311\times 10^{-5}$ for $K_{p},\;K_{d},\;and\;K_{i},$ respectively. The final prediction error (FPE) and MSE of the identified system are calculated as $1.33\times 10^{-4}\;and\;1.321\times 10^{-4}$.

In the experimental setup of the CTS, the physical input u is the $Q_{in}$ flow rate to the first tank controlled by the pump, which is actuated by a PWM signal generated by the MCU. Therefore, it is vital to express the mapping between PWM pulses that drive the pump motor and the input flow rate $Q_{in}$ of the CTS for implementing FL. The dataset covers the PWM ranges (from 0.0% to 100.0%) and the corresponding flow rate $\left(m^{3}/s\right)$ and includes real experimental data collected via the electromagnetic flow meter (Figure 14). The Matlab Curve Fitting Tool is employed to validate candidate models, which are obtained as follows: (i) 3rd degree polynomial with an RMSE value of 3.534, (ii) 2nd degree polynomial with an RMSE value of 3.553, and (iii) 2nd degree polynomial with bisquare fitting, resulting in a an RMSE value of 0.8424. The polynomial with the lowest RMSE is chosen to convert the physical input u into the corresponding PWM ratio and is given in Equation (5), where $x=Q_{in}$ denotes the input u and $f\left(x\right)$ denotes the corresponding PWM percentage.(5)$$f\left(x\right)=PWM(\%)=0.0003091x^{2}+0.00658x+22.38$$

### 2.10. FL-Based FLC

In the FL-based FLC design, inputs are denoted as $e_{k}=ref-{h_{2}}_{k}$ and $e_{k}^{\prime }=\frac{e_{k}-e_{k-1}}{T_{s}}$. The FLC design is explicitly described in Section 2.7. The crisp output of the FLC block is the new input v for the FL process. The actual control input u is calculated by Equation (4) and then applied to the nonlinear CTS simulated model or real experimental setup.

The overall control cycle algorithm, a so-called software flow chart embedded in the MCU electronics card-based system, can be demonstrated in Figure 15. Herein, all designed controller algorithms are coded sub-programs such as FLC, FL-based PID, and FL-based FLC.

## 3. Results

CTS simulation studies and experimental results for FLC, FL-based PID controller, and FL-based FLC methods are implemented under various input conditions. All evaluations and comparisons are based on performance index metrics such as RMSE, MAE, MSE, overshoot, and settling time.

### 3.1. CTS Simulation Setup

The designed CTS is modeled and simulated in Matlab/Simulink R2022b environment (Figure 16). Tank volumes, diameters, water density, temperature, and the acceleration of gravity are precisely calculated and selected from the experimental setup to create a digital twin model for simulation studies. Although tank levels are positive, the difference between the levels of tanks may be either positive or negative. Therefore, “signed square root” block is used to compute $\sqrt{h_{1}-h_{2}}\;$ flow gain. The system parameters are selected to be equal to the values obtained from the physical model. The maximum flow rate is limited to 450 l/h to ensure that it does not exceed the physical capacity of the inlet pump. Dimensionally, both tanks are equivalent to each other. The heights of the tanks, $h_{1}\;and\;h_{2}$, are 60 cm. The outer diameter $\left(D_{o}\right)$, the inner diameter $\left(D_{i}\right)$, and the wall thickness of the tanks are 9 cm, 8.3 cm, and 0.35 cm, respectively. The area of the water surface inside the cylindrical tank is $5411\;{mm}^{2}$, and the flow cross-section between the two tanks is $66.48\;{mm}^{2}$. The outlet cross-section used for fast discharge is also defined as $66.48\;{mm}^{2}$. The outlet water cross-section was narrowed using a servo motor-controlled ball valve and set to 22.28 mm^2^ (approximately one-third of the cross-section between the tanks). During the tests, the flow coefficients are defined as 0 for the discharge flow and 1 for the flow between the tanks and at the outlet cross-section. The sampling period for measurement and control signals is set to 0.2 s. The total test duration is 300 s, and during the tests, the gravitational acceleration is taken as $9.8066\;m/s^{2}$ to match the actual physical conditions.

#### 3.1.1. FLC Simulation Results

Four different set points (SPs) including two variable steps and two sinusoidal references are applied to the simulated FLC (Figure 17). The RMSE values for the FLC simulation are 0.06593 and 0.03992 under the reference step inputs shown in Figure 18a and Figure 18b, respectively. SP2 (Figure 18b) achieves a lower RMSE than SP1 (Figure 18a), as the step differences in SP2 are smaller. Overshoots are measured as 0.0087 m for SP1 and 0.0113 m for SP2, as shown in Figure 18a,b. SP2 also demonstrates a faster settling time of 37.2 s compared to SP1, attributed to the smaller step differences. For sinusoidal inputs, SP4 (Figure 18d) achieves a superior RMSE of 0.00689 compared to 0.01383 for SP3 (Figure 18c). In SP4, where the frequency components are lower than SP3, both the error and overshoot values are reduced compared to SP3, as illustrated in Figure 18c. The performance index metrics for the simulated FLC under each set point condition are given in Table 3.

#### 3.1.2. FL-Based PID Controller Simulation Results

The FL model of the CTS is simulated and controlled using a conventional PID controller (Figure 19). The PID parameters are determined via system identification and tuning tools in Matlab, as explained in Section 2.9. The simulated FL-based PID controller is sequentially tested with the same SPs, comprising two variable steps and two sinusoidal steps. Control signals and the controlled water level of the CTS are depicted in Figure 20a–c and d for SP1, SP2, SP3, and SP4, respectively. It is evident that zero overshoot is observed for both SP1 and SP2, while RMSE values diverge as 0.06161 and 0.04274, as presented in Figure 20a,b. The lower range steps in SP2 emphasize the advantage of a shorter settling time, calculated as 59.6 s, compared to 75.2 s for SP1. Concerning the tracking performance based on two sinusoidal inputs, SP4 in Figure 20d provides a better RMSE of 0.00691 than SP3, obtained as 0.01355 in Figure 20c. Due to its low-frequency components, both error and overshoot values obtained with SP4 are lower than those with SP3, as expected. The overall performances of the FL-based PID controller are given in Table 4.

#### 3.1.3. FL-Based FLC Simulation Results

The same four SPs are applied with FL-based FLC sequentially (Figure 21), with the controller signals and system responses for each input shown in Figure 22a–d. SP2 and SP4 input conditions yield superior results compared to SP1 and SP3, considering tracking errors and settling time. The RMSE values are computed as 0.059 and 0.03698 for SP1 and SP2, respectively, as presented in Figure 22a,b. Notably, there is no significant overshoot in the simulation of FL-based FLC for both SP1 and SP2 input conditions. A shorter settling time of 29.2 s is achieved for SP2, compared to 51.8 s for SP1. Regarding sinusoidal inputs, SP4 results in a better RMSE of $2.918\times 10^{-3}$ than SP3, which is obtained as $5.43\times 10^{-3}$ in Figure 22c. Throughout the level tracking process, both error and overshoot values are lower for SP4, given in Figure 22d, when compared to SP3. Table 5 shows the performances of simulated FL-based FLC in terms of overshoot, settling time, and tracking errors.

#### 3.1.4. Comparison of Simulation Results

Simulation results for the FLC, FL-based PID controller, and FL-based FLC are analyzed in Table 6, where the comparisons of these controller simulations in terms of tracking errors metrics are presented. The RMSE values of the FLC and FL-based PID controller are 0.0659 and 0.0616, respectively, higher than that of the FL-based FLC, which achieves the lowest RMSE of 0.059 under SP1 reference input conditions. For the variable step input SP2, the FL-based FLC similarly attains the best RMSE of 0.0370. However, a performance shift is observed between the FLC and FL-based PID controller, with FLC outperforming the FL-based PID controller by achieving a lower RMSE of 0.0399 compared to 0.0427. When responding to sinusoidal reference inputs SP3 and SP4, FLC and FL-based PID controllers demonstrate comparable performance in terms of RMSE, MAE, and MSE. As indicated, both controllers yield the same RMSE of 0.0069 for SP4, while their RMSE values for SP3 differ slightly at 0.0138 for the FLC and 0.0136 for the FL-based PID controller. The FL-based FLC achieves superior performance with RMSE, MAE, MSE values of $0.0054,\;0.0044,\;2.9\times 10^{-5}$ for SP3 and $0.0029,\;0.0017,\;8.5\times 10^{-6}$ for SP4, respectively, outperforming the other two controllers.

The simulated performances of the controllers are compared in Table 7, focusing on maximum overshoot and settling time. The results show that both FL-based approaches achieve zero overshoot for SP1 and SP2 reference inputs in FL-based PID controller and FL-based FLC simulations, whereas the FLC exhibits noticeable overshoot values of 0.0087 m and 0.0113 m for SP1 and SP2, respectively. However, for sinusoidal inputs SP3 and SP4, the FL-based PID controller experiences higher overshoot values of 0.0232 m and 0.0083 m compared to the other two approaches, which yield overshoot values of 0.0095 m and 0.0077 m in the FLC and followed by 0.0073 m and 0.0023 m in the FL-based FLC, respectively. In terms of settling time, the FLC outperforms the FL-based PID controller, achieving shorter settling times of 51.8 s and 29.2 s for SP1 and SP2 step inputs, along with lower overshoot values for sinusoidal inputs SP3 and SP4. Overall, the FL-based FLC demonstrates the best performance among the three controllers, achieving the shortest settling time and the lowest overshoot across all input conditions (SP1, SP2, SP3, and SP4) presented in Table 7.

### 3.2. Experimental Results for the Implemented CTS

The implemented real CTS setup is tested with the same SPs with FLC, FL-based PID, and FL-based FLC controllers. All performance results are presented and compared with each other in terms of RMSE, MAE, MSE, overshoot, and settling time.

#### 3.2.1. FLC Experimental Results

In the study conducted for the simulation of the FLC-controlled CTS presented in Figure 17, the actual CTS system was implemented instead of the SISO CTS model. The same four SPs are respectively given as SP1, SP2, SP3, and SP4 (Figure 23a–d). The RMSE values for the experimental FLC are determined as 0.06509 and 0.03956 under step values as SP1 and SP2 in Figure 23a and Figure 23b, respectively. Notable fluctuations in the control signal are evident when water levels are below 0.2 m in Figure 23a. A lower RMSE value is achieved in SP2 (Figure 23b), where the difference between step values is less than in SP1 (Figure 23a). The overshoots are observed as 0.011 m and 0.021 m for SP1 and SP2, respectively. The settling time performance in SP2, with 28.6 s, is better than in SP1 due to the smaller differences in step values. Concerning sinusoidal inputs, SP4 in Figure 23d yields a superior RMSE of 0.00712 compared to SP3, which is obtained as 0.01423 in Figure 23c. In SP4, where the frequency components are lower than SP3, both error and overshoot values are reduced. The real-time FLC performance results for SP1, SP2, SP3, and SP4 are detailed in Table 8.

#### 3.2.2. FL-Based PID Controller Experimental Results

The simulation model for the FL-based PID controller, as shown in Figure 19, has been implemented and applied to the physical CTS system for real-world testing and evaluation. The PID parameters are determined with detail in Section 2.9. Screenshots of the GUI are respectively given for the four SPs (namely SP1, SP2, SP3, and SP4) on the FL-based PID controller Figure 24a–d. The RMSE values for the real-time FL-based PID controller are calculated as 0.06722 and 0.04452 for the reference step inputs as SP1 and SP2 in Figure 24a and Figure 24b, respectively. Notably, there is zero overshoot observed for both of them. In Figure 24b, a settling time of 55.8 s is achieved for SP2, faster than the 72.4 s settling time for SP1 in Figure 24a. Regarding sinusoidal inputs, SP4 in Figure 24d yields a superior RMSE of 0.00476 compared to SP3, where the RMSE is 0.01249 in Figure 24c. Both error and overshoot values are lower for SP4 in Figure 24d when compared to SP3. The real-time FL-based PID controller performances are given in Table 9.

#### 3.2.3. FL-Based FLC Experimental Results

The simulation model of the FLC-based FLC, illustrated in Figure 21, has been deployed and tested on the physical CTS system for practical evaluation and analysis. The FLC inputs are calculated by the MCU, and the output of the FLC is scaled and then transmitted to the FL block as the linearized input v to obtain the nonlinear control signal u for the CTS as defined in Section 2.10. Screenshots of the GUI after experiments with SP1, SP2, SP3, and SP4 references on the FL-based FLC are respectively given in Figure 25a–d. RMSE values for the real-time FL-based FLC controller are computed as 0.06531 and 0.04406 for the reference step inputs as SP1 and SP2 in Figure 25a and Figure 25b, respectively. Although an insignificant overshoot of 0.003 m is recorded for SP1 in Figure 25a, which is still within the acceptable range of level measurement accuracy, SP2 exhibits zero overshoot. The settling time for SP2 is 24.8 s in Figure 25b, faster than the 54 s settling time for SP1 in Figure 25a. Regarding sinusoidal inputs, SP4 in Figure 25d yields a better RMSE of $4.02\times 10^{-3}$ compared to SP3, which is obtained as $9.31\times 10^{-3}$ in Figure 25c. Both error and overshoot values are lower for SP4 in Figure 25d when compared to SP3. The real-time performances of FL-based FLC are given in Table 10.

#### 3.2.4. Comparison of Experimental Results

The real-time performances of the FLC, FL-based PID, and FL-based FLC are evaluated with tracking error metrics in Table 11. For the SP1 step input, the FLC and FL-based FLC demonstrate similar RMSE values of 0.0651 and 0.0653, respectively, both outperforming the FL-based PID controller, which records an RMSE of 0.0672. However, for the SP2 step input, the FLC achieves the lowest RMSE of 0.0396, outperforming both the FL-based PID controller and the FL-based FLC whose RMSE values are as 0.0445 and 0.0441, respectively. Regarding the tracking performance under sinusoidal input conditions SP3 and SP4, Table 11 highlights significant differences in error metrics among the three candidate controllers, with the FL-based FLC achieving the best results, obtaining RMSE, MAE, and MSE values of $0.0093,\;0.0075,\;8.7\times 10^{-5}$ for SP3 and $0.0040,\;0.0028,1.6\times 10^{-5}$ for SP4, respectively. The FL-based PID controller’s performance ranks second, with resulting RMSE, MAE, and MSE values of $0.0125,\;0.0098,\;1.6{\times 10}^{-4}$ for SP3$\;0.0047\;and\;0.0032,\;2.3{\times 10}^{-5}$ for SP4, respectively, while FLC is outperformed due to the highest RMSE, MAE, and MSE values of $0.0142,\;0.0111,\;2.0{\times 10}^{-4}$ for SP3$\;and\;0.0071,\;0.0064,\;5.1{\times 10}^{-5}$ for SP4, respectively, making it the least effective in this comparison.

Table 12 presents a comparison of real-time controller performance in terms of maximum overshoot and settling time under different reference input conditions (SP1, SP2, SP3, and SP4). In terms of settling time, the FLC outperforms the FL-based PID controller, achieving 64.2 s and 28.6 s for SP1 and SP2 step inputs, compared to 72.4 s and 55.8 s for the FL-based PID controller. However, the FLC exhibits significant overshoots of 0.011 m and 0.021 m , whereas the FL-based PID controller achieves zero overshoot. Conversely, the FL-based FLC exhibits the best performance compared to both the FLC and the FL-based PID controller with shorter settling times of 54 s and 24.8 s under both SP1 and SP2 input conditions. Regarding overshoot, the FL-based PID performs slightly better than the FL-based FLC, particularly for SP1, with compared overshoot values of zero and 0.003 m, respectively. For the sinusoidal inputs SP3 and SP4, Table 12 illustrates overshoot performances where FL-based FLC attains a minimum overshoot of 0.012 m and 0.004 m. In this context, FLC and FL-based PID controllers exhibit similar overshoot values, such as 0.014 m, 0.013 m for SP3, and 0.009 m, 0.008 m for SP4, respectively. Overall, the FL-based FLC demonstrates superior performance in terms of both overshoot and settling time compared to FLC and FL-based PID controllers.

## 4. Conclusions

Three distinct nonlinear controllers were developed and implemented in an embedded ARM Cortex MCU with C codes for the level control of a physical CTS for a comprehensive comparative analysis: FLC, FL-based PID, and FL-based FLC. These controllers were evaluated under four different reference input conditions, including two step inputs and two sinusoidal inputs, to assess their tracking performance, stability, and robustness. These controllers demonstrated their capability to control on a designed microcontroller electronics card the inherently nonlinear CTS, and their performances were compared in terms of overshoot reduction and tracking accuracy. The simulations were done in Matlab/Simulink. However, real-time controllers are implemented on an MCU board without any support from the external control program Matlab/Simulink. For experimental studies, Matlab is used only for calculating controller performances from the real-time results. Digitally calibrated electromagnetic flow meters are used for flow measurement, which are guaranteed to an accuracy level of “less than 0.5% error” within the CTS flow range. Pump PWM values and the resulting flow rates are experimentally implemented from zero flow to maximum flow, and the mathematical model is created based on real-time mapping. Tank volumes, diameters, water density, temperature, and the acceleration of gravity are precisely calculated and selected to create a digital twin model for simulation studies.

The application of FL-based control strategies, such as the FL-based PID controller and the FL-based FLC, significantly contributed to performance improvements. In particular, it was observed that overshoot values, which initially ranged between 0.01 and 0.02 m under the FLC, were drastically reduced to 0.003 m and below with the introduction of FL-based methods under various reference step input conditions. This highlights the effectiveness of FL in mitigating large transient deviations, ensuring a smoother response in nonlinear systems. Furthermore, the FL-based control methods resulted in enhanced tracking performance, which is evident in the RMSE, MAE, and MSE values. For instance, under sinusoidal input condition SP3, the RMSE values for the FLC, FL-based PID, and FL-based FLC were computed as 0.0142, 0.0125, and 0.0093, respectively. Under the same input conditions, MAE and MSE values for the FL-based FLC were computed as 0.0349 and 0.0043, which are the minimum values obtained in comparison with FLC and FL-based PID controllers. A similar trend was observed with sinusoidal input SP4 when comparing the tracking performance of three controllers. The RMSE values for the controllers were recorded as 0.0071, 0.0047, and 0.0040, respectively, demonstrating higher levels of accuracy for the FL-based control strategy. The MAE values for these controllers were recorded as 0.0064, 0.0032, and 0.0028, further illustrating the differences in performance. Additionally, the MSE values were recorded as $5.1\times 10^{-5},\;2.3\times 10^{-5},\;and\;1.6\times 10^{-5}$, providing further insight into the FL-based FLC’s ability to minimize error under sinusoidal input conditions. These results reinforce the conclusion that FL-based FLC exhibits superior tracking performance, particularly in minimizing steady-state errors. Moreover, one notable observation in the study was the increase in settling time when using the FL-based PID controller. Specifically, under step input conditions, the settling time for the FL-based PID was recorded as 72.4 s, which is longer compared to the FLC settling time of 64.2 s. Conversely, the FL-based FLC demonstrated the best settling time performance, reducing it to 54 s, making it the most effective among the three approaches in terms of response speed and overall efficiency.

As for the future research directions, several research avenues can be explored to further optimize system performance: (i) Future research can focus on implementing adaptive control techniques, where controller parameters are dynamically adjusted in real time to account for system uncertainties and disturbances. Additionally, robust control strategies such as optimal control or sliding mode control can be integrated to improve system resilience against parameter variations. (ii) The incorporation of reinforcement learning and neural network-based controllers can be investigated to develop intelligent control strategies that automatically adapt to nonlinearities without requiring explicit mathematical modeling. (iii) The possible study can be extended to multi-input multi-output systems, where multiple CTS or interconnected processes require coordinated control strategies. (iv) Implementing low-power embedded control strategies for real-time applications can be another area of investigation. The computational efficiency of advanced controllers can be further optimized for embedded microcontroller platforms, making them more suitable for industrial automation and IoT-based monitoring systems.

## Figures and Tables

**Figure 1 sensors-25-01279-f001:**
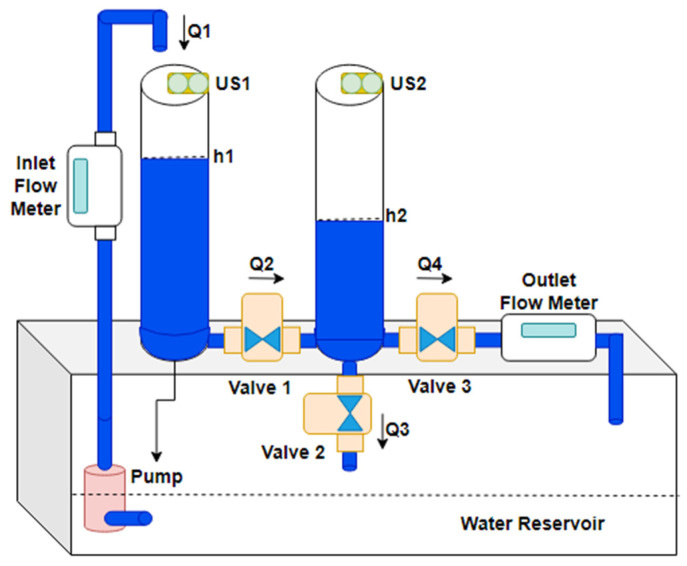
Closed loop flow diagram of CTS.

**Figure 2 sensors-25-01279-f002:**
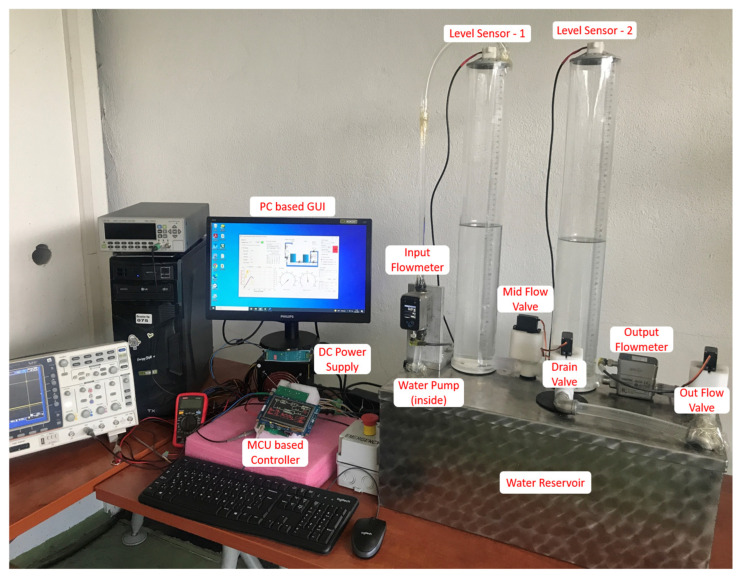
Layout of implemented CTS.

**Figure 3 sensors-25-01279-f003:**
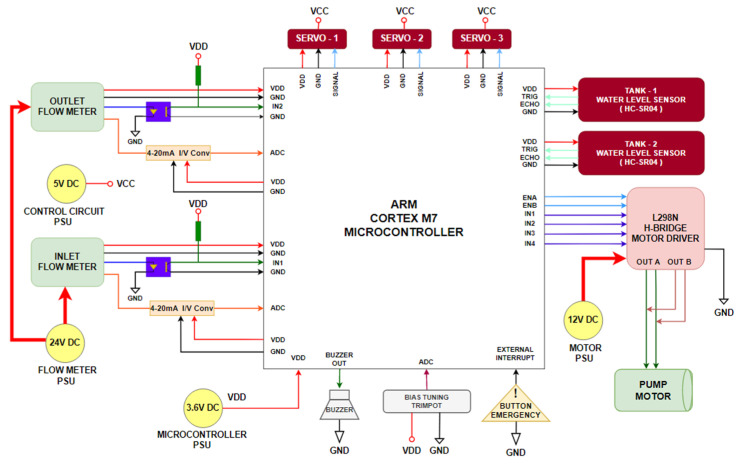
Electrical block diagram of CTS.

**Figure 4 sensors-25-01279-f004:**
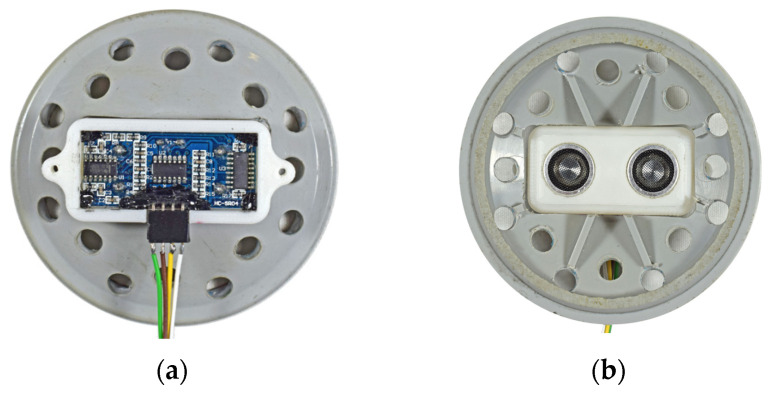
HC-SR04 ultrasonic level sensor assembled in a specially designed 3D printed case for installation on top of each tank. (**a**) Top view of the sensor assembly with the protection cover open; (**b**) Bottom view of the sensor with transmitter and receiver, facing the water surface in the tank.

**Figure 5 sensors-25-01279-f005:**
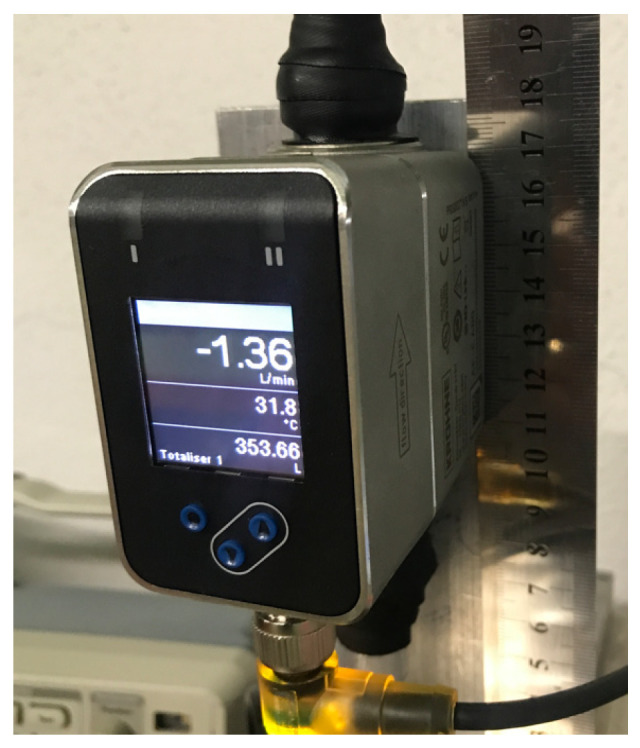
AF-E-400 electromagnetic flow meter.

**Figure 6 sensors-25-01279-f006:**
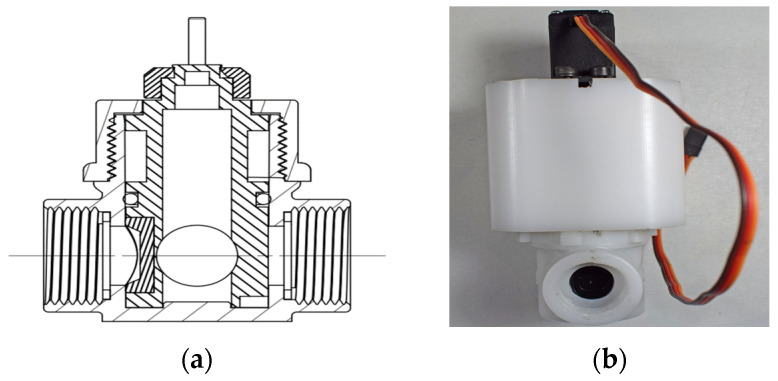
Specially designed ball valve assembly and servo motor. (**a**) Technical drawing of designed ball valve. (**b**) Front view of ball valve assembled in a 3D-printed casing under MG-996R servo motor.

**Figure 7 sensors-25-01279-f007:**
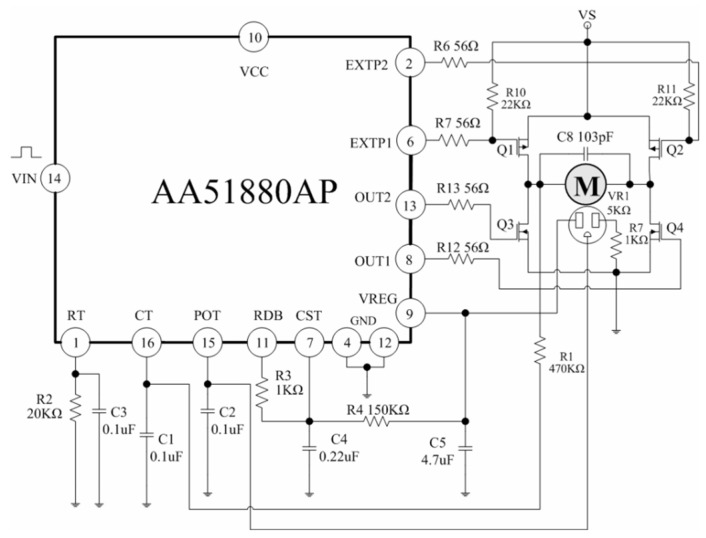
The electrical schematics of MG996R servo motor controller.

**Figure 8 sensors-25-01279-f008:**
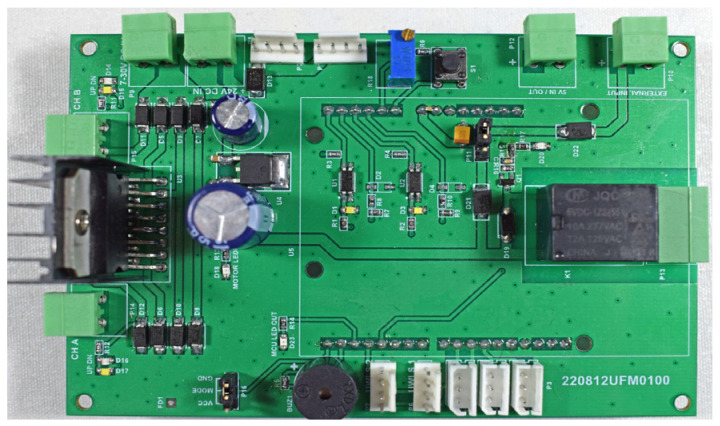
The implemented electronic interface card designed on STM32F746 evaluation board.

**Figure 9 sensors-25-01279-f009:**
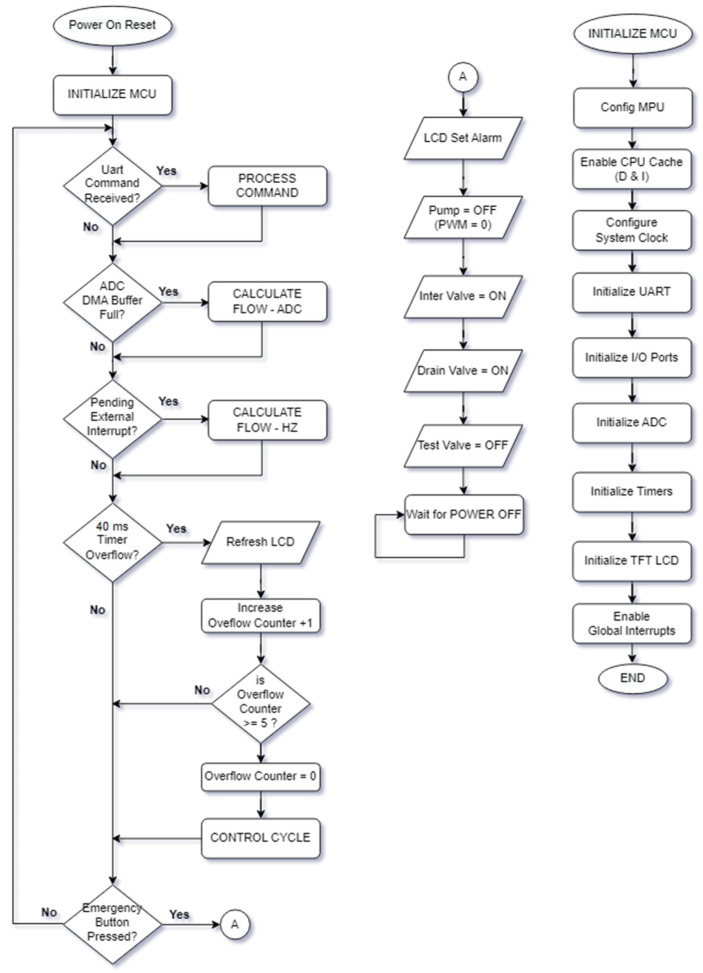
The designed embedded software flow chart.

**Figure 10 sensors-25-01279-f010:**
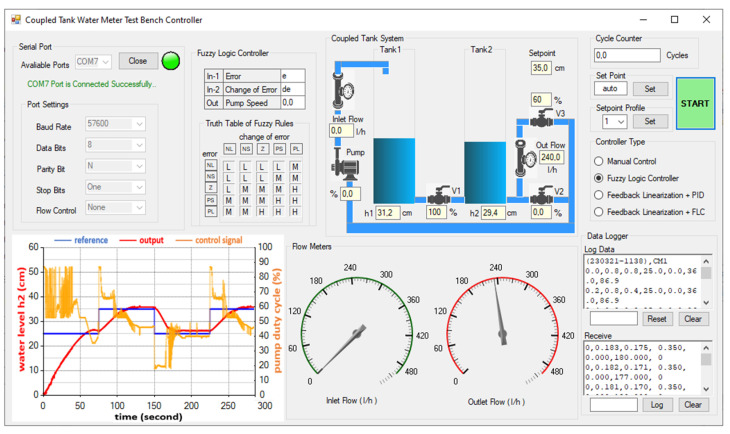
The designed GUI possessing control and observation panels.

**Figure 11 sensors-25-01279-f011:**
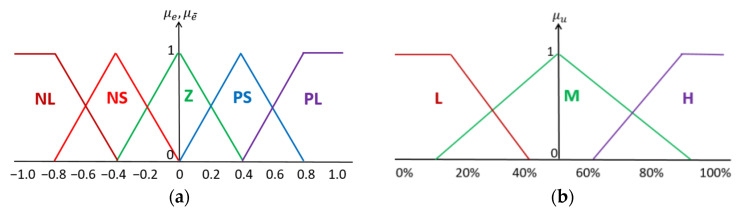
Membership functions: (**a**) error and change-in-error inputs; (**b**) motor PWM output.

**Figure 12 sensors-25-01279-f012:**
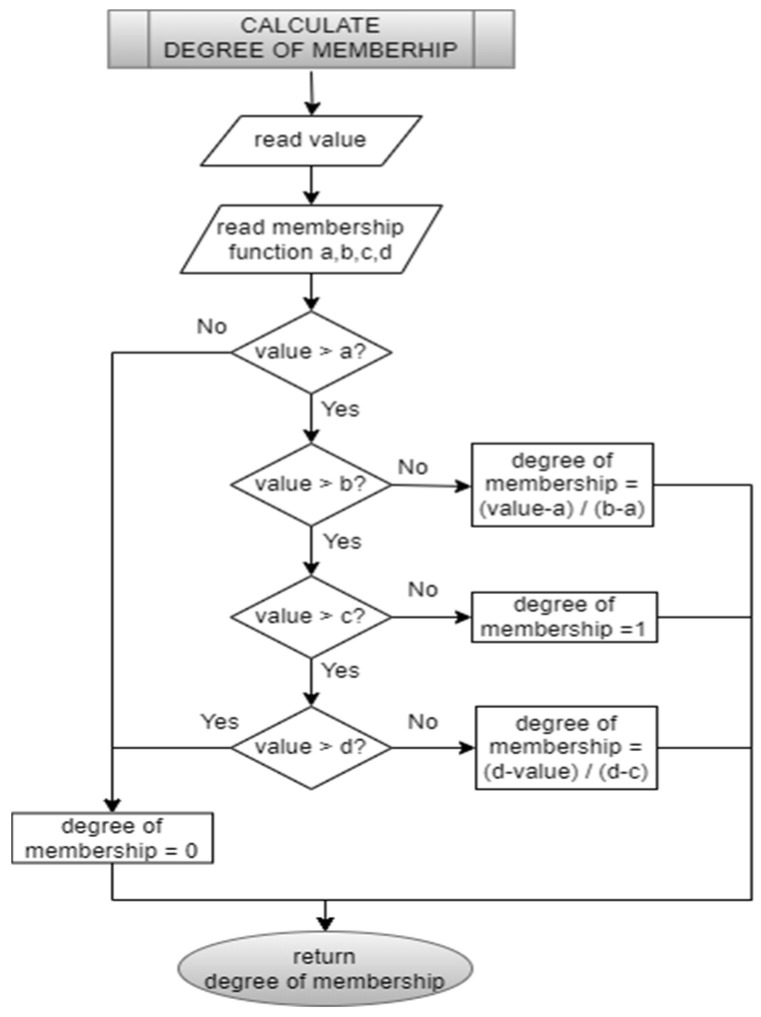
Degree-of-membership calculation flow chart.

**Figure 13 sensors-25-01279-f013:**
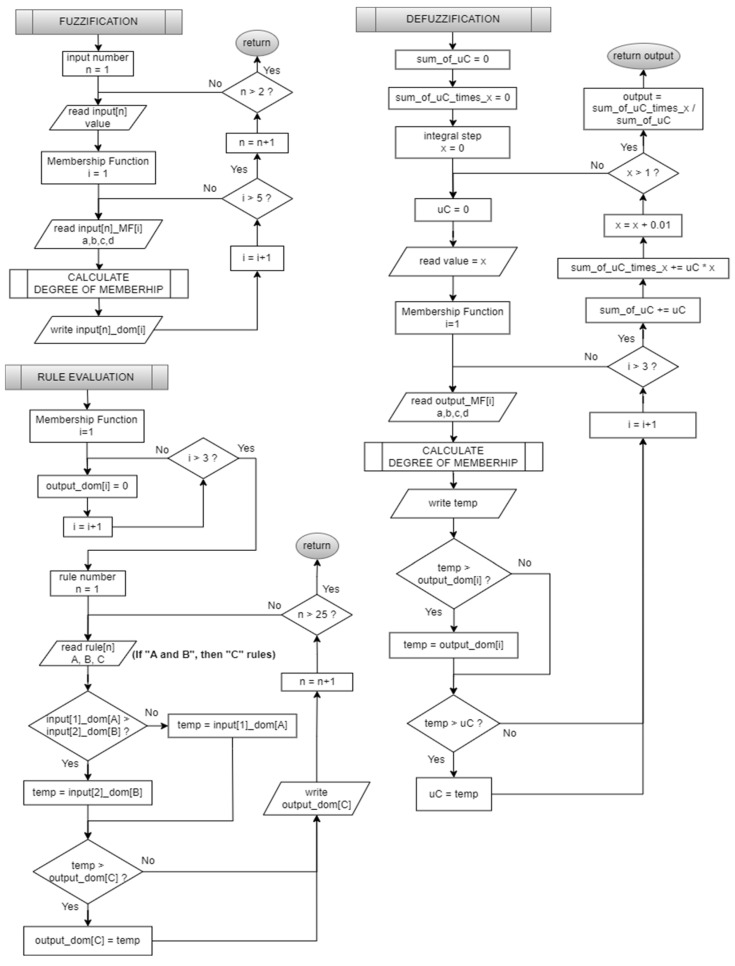
FLC stages flow chart.

**Figure 14 sensors-25-01279-f014:**
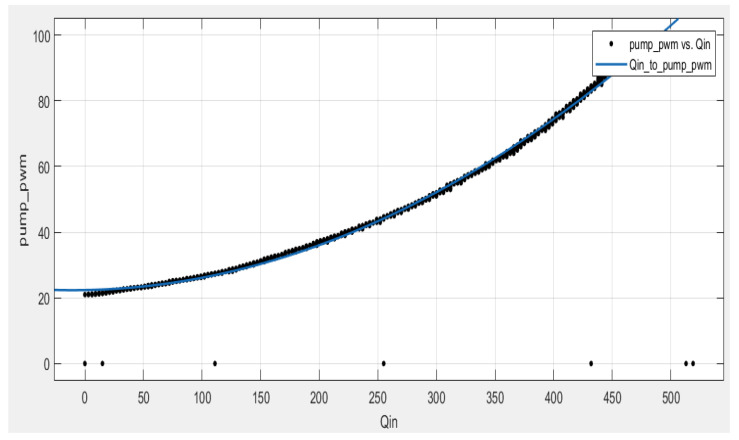
Curve fitting plot for input flow rate to PWM conversion.

**Figure 15 sensors-25-01279-f015:**
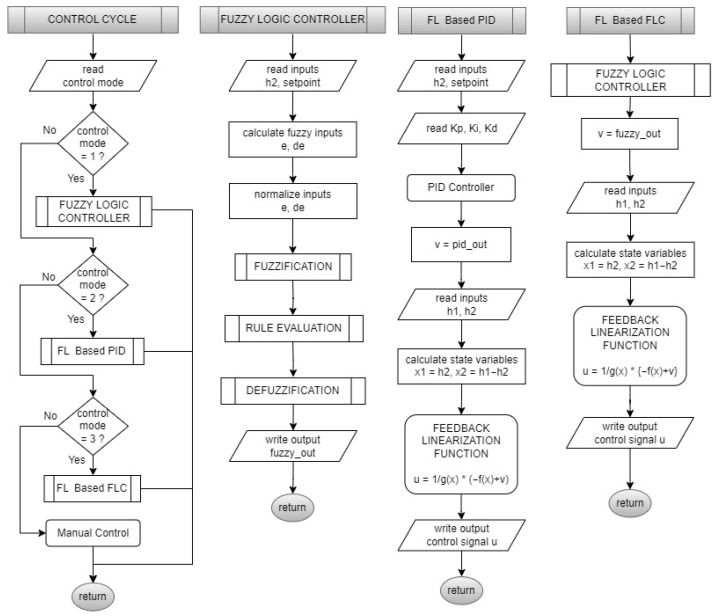
Flow charts of implemented control cycle, FLC, FL-based PID, and FL-based FLC.

**Figure 16 sensors-25-01279-f016:**
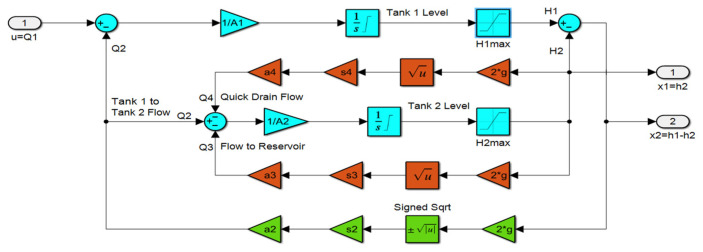
SISO model of CTS.

**Figure 17 sensors-25-01279-f017:**
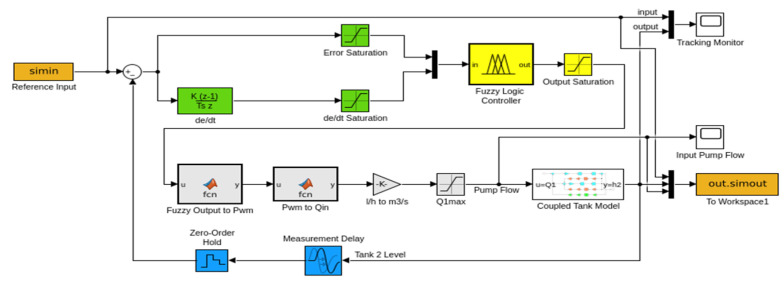
FLC control on CTS model.

**Figure 18 sensors-25-01279-f018:**
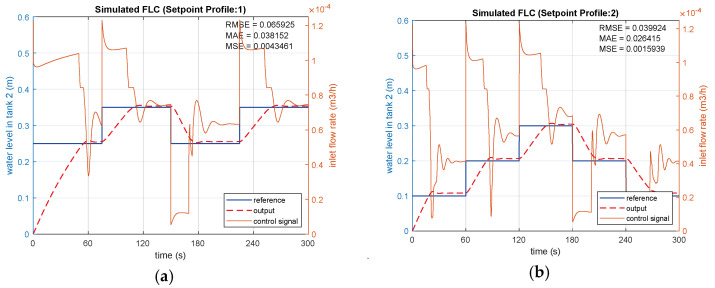
Simulations of FLC with (**a**) SP1; (**b**) SP2; (**c**) SP3; (**d**) SP4.

**Figure 19 sensors-25-01279-f019:**
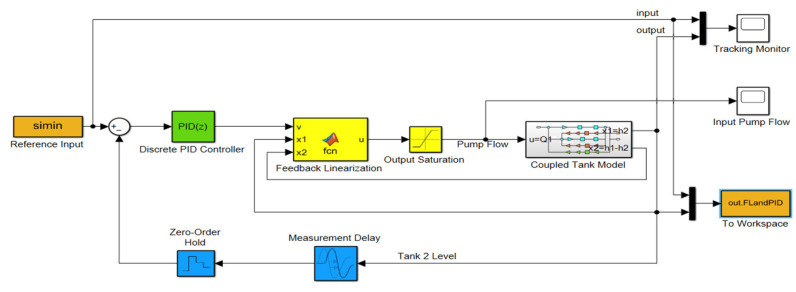
FL-based PID controller model.

**Figure 20 sensors-25-01279-f020:**
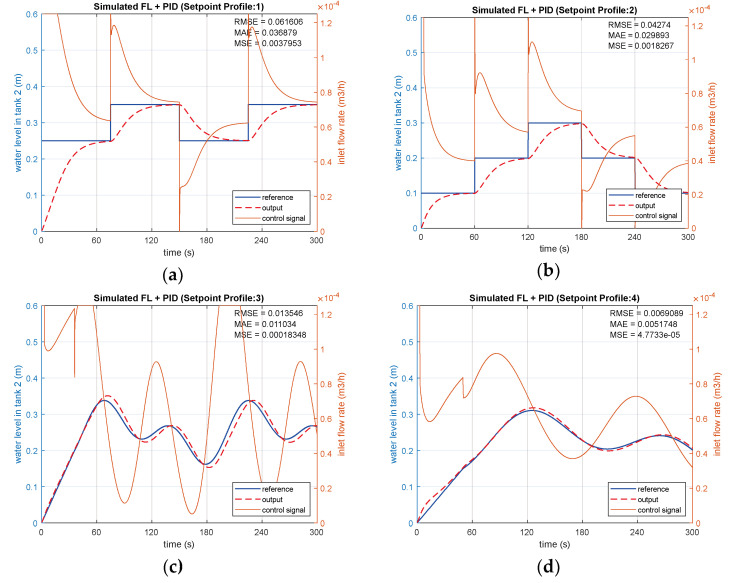
Simulations of FL-based PID controller with (**a**) SP1; (**b**) SP2; (**c**) SP3; (**d**) SP4.

**Figure 21 sensors-25-01279-f021:**
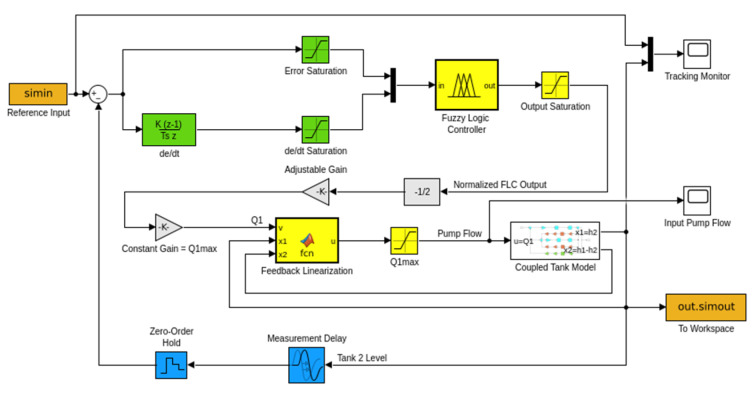
FL-based FLC model.

**Figure 22 sensors-25-01279-f022:**
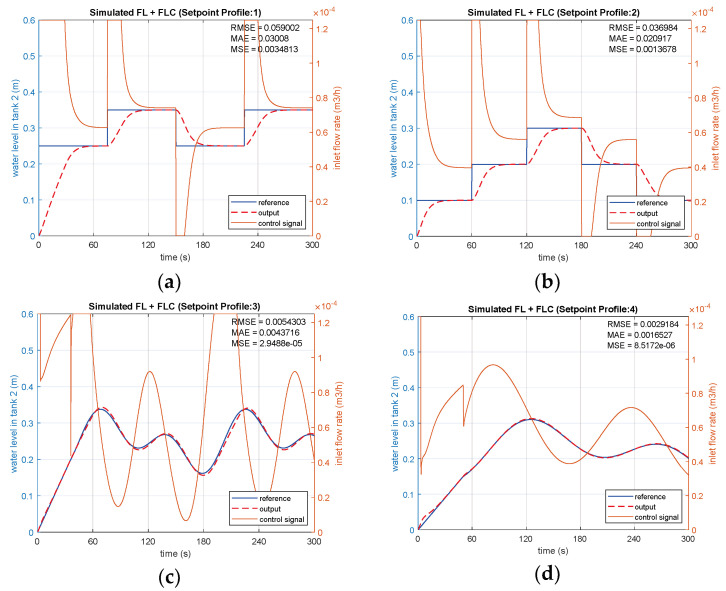
Simulations of FL-based FLC with (**a**) SP1; (**b**) SP2; (**c**) SP3; (**d**) SP4.

**Figure 23 sensors-25-01279-f023:**
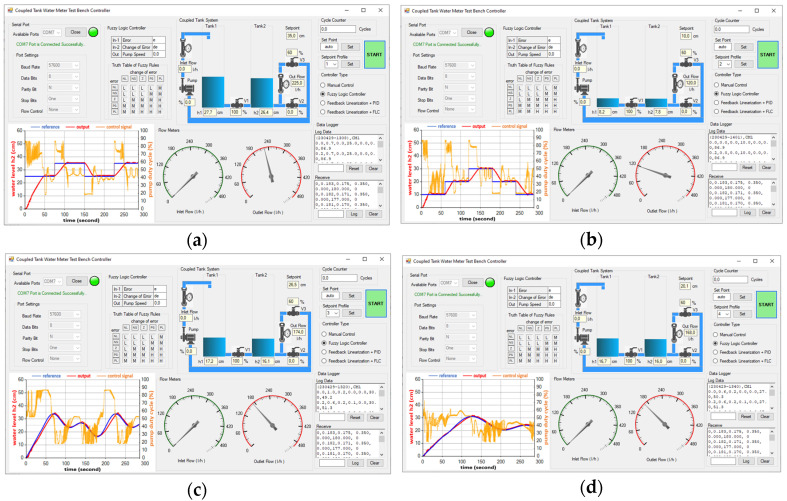
Test screenshots from GUI for FLC with (**a**) SP1; (**b**) SP2; (**c**) SP3; (**d**) SP4.

**Figure 24 sensors-25-01279-f024:**
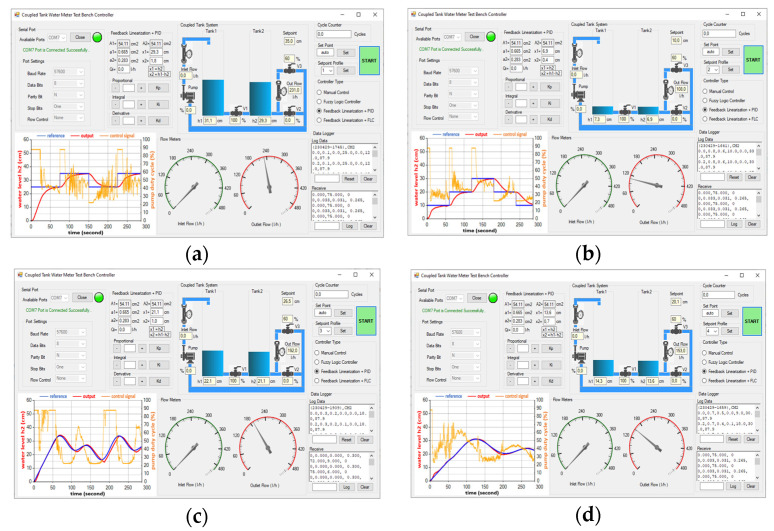
Test screenshots from GUI for FL-based PID controller with (**a**) SP1; (**b**) SP2; (**c**) SP3; (**d**) SP4.

**Figure 25 sensors-25-01279-f025:**
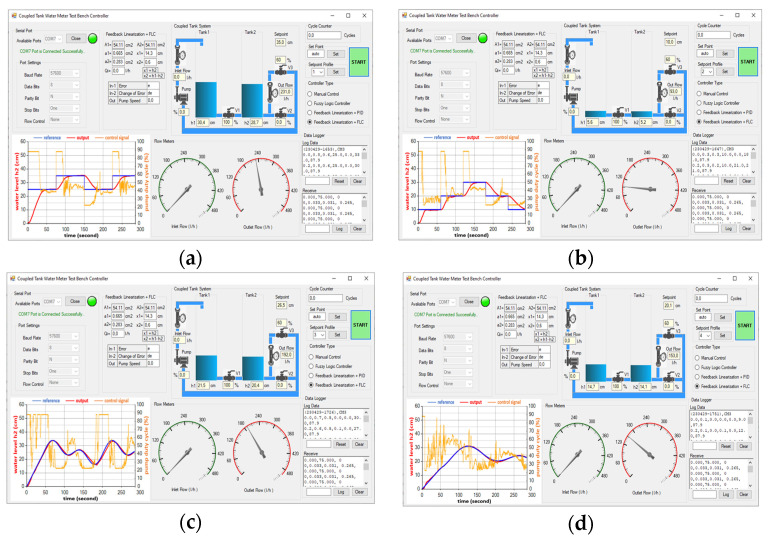
Test screenshots from GUI for FL-based FLC with (**a**) SP1; (**b**) SP2; (**c**) SP3; (**d**) SP4.

**Table 1 sensors-25-01279-t001:** GUI commands and parameters used for UART serial communication with CTS.

Command	Parameter 1	Parameter 2	Parameter 3	Explanation
Write (W)	P	1	pwm	set pump driver pwm ratio
	V	1	valve-1	set valve-1 orifice ratio
		2	valve-2	set valve-2 orifice ratio
		3	valve-3	set valve-3 orifice ratio
	S	1	set point	write constant setpoint (step)
	Kp	PID-Kp	-	write PID coefficient Kp
	Ki	PID-Ki	-	write PID coefficient Ki
	Kd	PID-Kd	-	write PID coefficient Kd
Read (R)	N	Cycle Cnt	-	read instantaneous control cycle
	P	pump	-	read pump pwm ratio
	L	levels	-	read both tank water levels
	V	valves	-	read all valve openings
Control Mode (CM)	0	manual	-	switch to manual control mode
	1	fuzzy	-	switch to fuzzy controller
	2	FL + PID	-	switch to FL + PID controller
	3	FL + fuzzy	-	switch to FL + fuzzy controller
Control (C)	B	0–1	-	buzzer on/off
	R	0–1	-	relay on/off
	E	0–1	-	emergency on/off
	P	0–1	-	pump main power on/off
	S	1–4	-	switch setpoint profile

**Table 2 sensors-25-01279-t002:** FLC fuzzy associative matrix for CTS.

e’		NL	NS	Z	PS	PL
	e					
NL		L	L	L	L	M
NS		L	L	L	M	M
Z		L	M	M	M	H
PS		M	M	M	H	H
PL		M	M	H	H	H

**Table 3 sensors-25-01279-t003:** Simulated FLC performances.

SP	Input Type	RMSE	MAE	MSE	Overshoot (m)	Settling Time (s)
SP1	step input–1	0.06593	0.03815	4.346 × 10^−3^	0.0087	71.8
SP2	step input–2	0.03992	0.02642	1.594 × 10^−3^	0.0113	37.2
SP3	sinusoidal input–1	0.01383	0.01081	1.912 × 10^−4^	0.0095	-
SP4	sinusoidal input–2	0.00689	0.00627	4.751 × 10^−5^	0.0077	-

**Table 4 sensors-25-01279-t004:** Simulated FL-based PID controller performances.

SP	Input Type	RMSE	MAE	MSE	Overshoot (m)	Settling Time (s)
SP1	step input–1	0.06161	0.03688	3.795 × 10^−3^	0	75.2
SP2	step input–2	0.04274	0.02989	1.827 × 10^−3^	0	59.6
SP3	sinusoidal input–1	0.01355	0.01103	1.835 × 10^−4^	0.0232	-
SP4	sinusoidal input–2	0.00691	0.00518	4.773 × 10^−5^	0.0083	-

**Table 5 sensors-25-01279-t005:** Simulated FL-based FLC performances.

SP	Input Type	RMSE	MAE	MSE	Overshoot (m)	Settling Time (s)
SP1	step input–1	0.059	0.03008	3.481 × 10^−3^	0	51.8
SP2	step input–2	0.03698	0.02092	1.368 × 10^−3^	0	29.2
SP3	sinusoidal input–1	5.43 × 10^−3^	4.372 × 10^−3^	2.949 × 10^−5^	0.0073	-
SP4	sinusoidal input–2	2.918 × 10^−3^	1.653 × 10^−3^	8.517 × 10^−6^	0.0023	-

**Table 6 sensors-25-01279-t006:** Comparisons of FLC, FL-based PID, and FL-based FLC simulations in terms of tracking errors.

	FLC			FL-Based PID			FL-Based FLC		
SP	RMSE	MAE	MSE	RMSE	MAE	MSE	RMSE	MAE	MSE
SP1	0.0659	0.0382	0.0043	0.0616	0.0369	0.0038	0.059	0.0301	0.0035
SP2	0.0399	0.0264	0.0016	0.0427	0.0299	0.0018	0.0370	0.0209	0.0014
SP3	0.0138	0.0108	1.9 × 10^−4^	0.0136	0.0110	1.8 × 10^−4^	0.0054	0.0044	2.9 × 10^−5^
SP4	0.0069	0.0063	4.8 × 10^−5^	0.0069	0.0052	4.8 × 10^−5^	0.0029	0.0017	8.5 × 10^−6^

**Table 7 sensors-25-01279-t007:** Comparison of simulation results in terms of overshoot and settling time.

	FLC		FL-Based PID		FL-Based FLC	
SP	Overshoot (m)	Settling Time (s)	Overshoot (m)	Settling Time (s)	Overshoot (m)	Settling Time (s)
SP1	0.0087	71.8	0	75.2	0	51.8
SP2	0.0113	37.2	0	59.6	0	29.2
SP3	0.0095	-	0.0232	-	0.0073	-
SP4	0.0077	-	0.0083	-	0.0023	-

**Table 8 sensors-25-01279-t008:** Experimental FLC performances.

SP	Input Type	RMSE	MAE	MSE	Overshoot (m)	Settling Time (s)
SP1	step input–1	0.06509	0.03559	4.236 × 10^−3^	0.011	64.2
SP2	step input–2	0.03956	0.02513	1.564 × 10^−3^	0.021	28.6
SP3	sinusoidal input–1	0.01423	0.01112	2.026 × 10^−4^	0.014	-
SP4	sinusoidal input–2	0.00712	0.00639	5.067 × 10^−5^	0.009	-

**Table 9 sensors-25-01279-t009:** Experimental FL-based PID controller performances.

SP	Input Type	RMSE	MAE	MSE	Overshoot (m)	Settling Time (s)
SP1	step input–1	0.06722	0.03976	4.518 × 10^−3^	0	72.4
SP2	step input–2	0.04452	0.03062	1.982 × 10^−3^	0	55.8
SP3	sinusoidal input–1	0.01249	0.00983	1.560 × 10^−4^	0.013	-
SP4	sinusoidal input–2	0.00476	0.00319	2.266 × 10^−5^	0.008	-

**Table 10 sensors-25-01279-t010:** Experimental FL-based fuzzy logic controller performance in terms of overshoot, settling time, and tracking errors.

SP	Input Type	RMSE	MAE	MSE	Overshoot (m)	Settling Time (s)
SP1	step input–1	0.06531	0.03487	4.265 × 10^−3^	0.003	54
SP2	step input–2	0.04406	0.0281	1.942 × 10^−3^	0	24.8
SP3	sinusoidal input–1	9.31 × 10^−3^	7.524 × 10^−3^	8.667 × 10^−5^	0.012	-
SP4	sinusoidal input–2	4.02 × 10^−3^	2.863 × 10^−3^	1.616 × 10^−5^	0.004	-

**Table 11 sensors-25-01279-t011:** Comparison of experimental results in terms of tracking errors.

	FLC			FL-Based PID			FL-Based FLC		
SP	RMSE	MAE	MSE	RMSE	MAE	MSE	RMSE	MAE	MSE
SP1	0.0651	0.0356	0.0042	0.0672	0.0398	0.0045	0.0653	0.0349	0.0043
SP2	0.0396	0.0251	0.0016	0.0445	0.0306	0.002	0.0441	0.0281	0.0019
SP3	0.0142	0.0111	2.0 × 10^−4^	0.0125	0.0098	1.6 × 10^−4^	0.0093	0.0075	8.7 × 10^−5^
SP4	0.0071	0.0064	5.1 × 10^−5^	0.0047	0.0032	2.3 × 10^−5^	0.0040	0.0028	1.6 × 10^−5^

**Table 12 sensors-25-01279-t012:** Comparison of experimental results in terms of overshoot and settling time.

	FLC		FL-Based PID		FL-Based FLC	
SP	Overshoot (m)	Settling Time (s)	Overshoot (m)	Settling Time (s)	Overshoot (m)	Settling Time (s)
SP1	0.011	64.2	0	72.4	0.003	54
SP2	0.021	28.6	0	55.8	0	24.8
SP3	0.014	-	0.013	-	0.012	-
SP4	0.009	-	0.008	-	0.004	-

## Data Availability

The original contributions presented in this study are included in the article. The raw data obtained and used for analysis are openly available in Github at https://github.com/bahadiryesil/sensors-25-01279.

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
