# Peer review of "Real-Time Implementation of a Microcontroller-Based Coupled-Tank Water Level Control System with Feedback Linearization and Fuzzy Logic Controller Algorithms"

_sensors, 2025, doi:10.3390/s25051279_

Round 1
Reviewer 1 Report
Comments and Suggestions for Authors
The research is based on application of linearisation technique using fuzzy logic to control coupled tanks level. It includes design of three types of controllers - FLC, FL-based PID controller, and FL-based FLC . Resulta from simulation and experimental study for different reference changes are widely discussed and compared.
My comments and recommendations are the following:
- Mamdani or Sugeno PID FLC is a typical example for control of nonlinear plants widely applied in laboratory or industrial practice with easy design and implementation. It is good to be included as a basis for comparison. The PID FLC is nonlinear and by correct design compensates the plant nonlinearity.
- A comparison with a classical linearising controller (approximation only of the linearising part) is also good
- Membership functions - orthogonal identical triangles ususlly makes the FLC linear. Then the question is why FL is used.
- In the conclusion section it is good to explain the vision of the authors for their future research.
The amount of work done is big and the presentation too detailed which hinders the understanding of the basic results. It is good to reduce the number of figures (now 32) and tables (now 17) and to integrate the discussions on the results for each case.
Author Response
Please see the attachment,
Best Regards

Reviewer 2 Report
Comments and Suggestions for Authors
Dear authors,
The authors elaborated on the real-time implementation of a microcontroller coupled water tank level control system based on feedback linearization and fuzzy logic controller algorithm. There are some details that need to be further supplemented.
Please find the attached file.

Author Response
Please see the attachment,
Best Regards.

Round 2
Reviewer 2 Report
Comments and Suggestions for Authors
Accept in present form.